# Higher CSF sTNFR1-related proteins associate with better prognosis in very early Alzheimer's disease

William T. Hu [ORCID] [1,74✉], Tugba Ozturk[1], Alexander Kollhoff[1], Whitney Wharton[2], J. Christina Howell[1], Alzheimer's Disease Neuroimaging Initiative*

Neuroinflammation is associated with Alzheimer's disease, but the application of cerebrospinal fluid measures of inflammatory proteins may be limited by overlapping pathways and relationships between them. In this work, we measure 15 cerebrospinal proteins related to microglial and T-cell functions, and show them to reproducibly form functionally-related groups within and across diagnostic categories in 382 participants from the Alzheimer's Disease Neuro-imaging Initiative as well participants from two independent cohorts. We further show higher levels of proteins related to soluble tumor necrosis factor receptor 1 are associated with reduced risk of conversion to dementia in the multi-centered ($p = 0.027$) and independent ($p = 0.038$) cohorts of people with mild cognitive impairment due to predicted Alzheimer's disease, while higher soluble TREM2 levels associated with slower decline in the dementia stage of Alzheimer's disease. These inflammatory proteins thus provide prognostic information independent of established Alzheimer's markers.

[1] Department of Neurology and Center for Neurodegenerative Diseases, School of Medicine, Emory University, Atlanta, GA, USA. [2] Nell Hodgson School of Nursing, Emory University, Atlanta, GA, USA. [74]Present address: Rutgers Robert Wood Johnson Medical School and Rutgers Institute for Health, Health Care Policy and Aging Research, Rutgers Biomedical and Health Sciences, New Brunswick, NJ, USA. *A list of authors and their affiliations appears at the end of the paper. ✉email: william.hu@rutgers.edu

The early and accurate detection of core Alzheimer's disease (AD) pathologies is increasingly probable with reliable cerebrospinal fluid (CSF) and molecular imaging biomarkers[1]. Longitudinal studies have consistently shown biomarkers for AD neuropathologic changes (ADNC)[2] to address IF people with the earliest cognitive symptoms will undergo cognitive decline, but they poorly address WHEN such decline will occur—especially at the individual level (Fig. S1). Major reasons for this include the relatively stable trajectory of CSF (amyloid and tau) or PET amyloid biomarkers once in the symptomatic stages of AD[3–6], clinical and pathologic AD heterogeneity, and mixed pathologies. Biomarkers reflecting biological processes commonly found in AD, but relatively independent of the formation of neuritic plaques and neurofibrillary tangles, may provide crucial information on the brain's susceptibility or resistance to ADNC.

Among these processes, neuroinflammation is consistently identified to relate to AD pathogenesis. Multiple genetic variants associated with increased AD risks were found in immune-related genes including *CD33*, *CR1*, *HLA-DRB5-HLA-DRB1*, *MEF2C*, *TREM2*, and *PLCG2*. Neuropathologic analysis has commonly shown microglial activation in AD, and we and others previously showed stage-specific CSF alterations in complement and interleukin levels[7–10]. Moreover, AD appears to specifically modify aging-related T-cell cytokine alterations (inflammaging)[11]. Thus, CSF inflammatory proteins and peptides represent promising candidates to inform rates of AD progression. In keeping with this, higher CSF soluble TREM2 (sTREM2) levels were recently linked to slower AD progression in the Alzheimer's Disease Neuroimaging Initiative (ADNI)[12]. This is of particular interest because loss-of-function mutations in the sTREM2 sheddace *TACE/ADAM17* are linked to AD[13], and other substrates for this enzyme are commonly implicated in the assessment of AD-related inflammatory changes[8,14–18].

Because CSF inflammatory proteins are regulated by interlinked pro- and anti-inflammatory processes, it is not straightforward to postulate how their levels vary according to each other and AD. Ideal CSF inflammatory biomarkers should also have readily available assays with high accuracy and intermediate precision, and CSF changes relatively orthogonal to core AD biomarkers beta-amyloid 1–42 (Aβ42), total tau (t-Tau), and tau phosphorylated at threonine 181 (p-Tau$_{181}$). To-date, we are only aware of one study which assessed multiple CSF inflammatory proteins as prognostic biomarkers in the Swedish BioFINDER study using a customized assay kit[10].

In this work, we analyze levels of 15 CSF inflammatory proteins implicated in microglial- and T-cell-mediated processes in the multi-centered ADNI (Table 1) blinded to diagnosis and prognosis, including soluble Tumor Necrosis Factor Receptor 1 and 2 (sTNFR1,2); Transforming Growth Factor 1, 2, and 3 (TGFβ 1,2,3); soluble Intercellular Adhesion Molecule 1 (sICAM1) and soluble Vascular Cell Adhesion Molecule 1 (sVCAM1); Tumor Necrosis Factor α (TNFα); Interleukin 6 (IL-6), IL-7, IL-9, IL-10, IL-12p40, IL-21; and Interferon Gamma-Induced Protein 10 (IP-10). Of these, sTNFR1, sTNFR2, sICAM1, sVCAM1, and TNFα are all substrates for TACE/ADAM17[8,14–18]. We identify consistently correlated families of proteins using principal component analysis (PCA) within each ADNI diagnostic category, and find sTNFR1-related proteins and sTREM2 to associate with prognosis[19] (rates and likelihood of decline) in mild cognitive impairment (MCI) and AD dementia in a diagnosis-specific manner.

## Results

### CSF inflammatory proteins form reproducible families across diagnostic categories.
ADNI participants included in the current study showed highly variable rates of longitudinal cognitive and functional decline even when classified by clinical diagnosis and predicted ADNC status (by t-Tau/Aβ42 ratio; Supplementary Fig. 1, Supplementary Table 1)[20]. To identify potential correlations between functionally-related CSF inflammatory proteins (after log-10 and Z-transformation; see "Methods" and Supplementary Tables 2 and 3), we performed dimension reduction using PCA[21] on the 15 proteins we measured with previously measured levels of core AD biomarkers (Aβ42, t-Tau, p-Tau$_{181}$, t-Tau/Aβ42), sTREM2, and progranulin. PCA was conducted independently within the normal cognition (NC), MCI, and AD dementia cohort to avoid false discovery due to contrast between extreme subgroups (NC vs moderate AD dementia). We found six highly reproducible principal components (PCs) within each diagnostic category (Table 2 and Supplementary Table 4): core AD, sTNFR1 (also sTNFR2, sVCAM1, sICAM1, TGFβ1), sTREM2, IL-6/IL-10, TGFβ (TGFβ2, TGFβ1), and IL-7/TNF-α, even though there were instances where some variables demonstrated diagnosis-specific loading (e.g., IP-10 on PC2 in the two groups with AD dementia). People with NC and MCI additionally shared another PC consisting of IP-10 and IL-12p40. PCA in two independent cohorts for whom core AD biomarkers and nine CSF inflammatory protein levels were previously and independently measured (Supplementary Methods; Table S5) replicated PCs for core AD, sTNFR1, IL-10, and IL-7/TNF-α PCs, but sTREM2 (measured using a commercially available assay distinct from the two used in ADNI) became a member of the sTNFR1 PC. These highly reproducible groupings provided strong empirical evidence for CSF inflammatory biomarker families or clusters.

### CSF AD and pro-inflammatory alterations each associated with 5-year cognitive decline in MCI.
Based on the inflammatory protein PCs' orthogonality from each other and from core AD biomarker PC, we tested if PC scores correlated with rates of longitudinal cognitive and functional decline. We began with MCI participants because they had longer follow-up duration than AD dementia and higher likelihood of decline than NC. Linear mixed modeling (LMM)—taking into account baseline characteristics (age, sex, self-reported race, *APOE* genotype) and CSF protein family PC scores—was paired with two measures of decline: average of composite ADNI Memory and Executive Function scores (ADNI-Mem-EF) and Clinical Dementia Rating Sum of Boxes (CDR-SB). Models which incorporated non-AD biomarker scores outperformed those which only incorporated core AD scores (Supplementary Tables 6, 7), with faster decline during 60 months following CSF collection independently associated with older age ($p < 0.001$), greater core AD score ($p < 0.001$), and lower sTNFR1 score levels ($p ≤ 0.002$, Table 3) for both outcomes.

We then tested whether specific biomarker PC scores can risk stratify within the MCI cohort. We first split the MCI group into those with high or low core AD biomarker scores according to the value ($-0.614$) corresponding to the t-Tau/Aβ42 threshold for ADNC[20]. At this core AD score threshold, a sTNFR1 score of $-0.890$ is expected to result in no net decline in ADNI-Mem-EF over time and is therefore used to further stratify MCI participants (Fig. 1A). We found participants with high core AD score and low sTNFR1 score to have the earliest decline (according to diagnostic conversion or CDR-SB) during the 60 months following CSF collection. Compared to this group, those with similar core AD scores but higher sTNFR1 scores were less likely to decline ($p = 0.014$ by consensus diagnosis, median time to conversion of 36 vs 12 months; $p = 0.007$ by CDR-SB ≥ 4, median time to conversion of 48 vs 24 months, Fig. 1B). Because dementia diagnosis and greater CDR-SB both imply functional

**Table 1 Subjects included in study from ADNI, with _p_ values shown for all continuous factors without log$_{10}$ transformation but also for CSF inflammatory proteins after log$_{10}$ transformation.**

| | NC (_n_ = 111) | MCI (_n_ = 174) | AD (_n_ = 97) | _P_ | _p_ for log$_{10}$ values |
|---|---|---|---|---|---|
| Male (%) | 57 (51%) | 112 (64%) | 56 (58%) | 0.090 | |
| Age, mean (SD) | 75.8 (5.3) | 75.2 (7.6) | 75.1 (7.8) | 0.659 | |
| Education, mean (SD) | 15.7 (2.9) | 15.8 (2.9) | 15.2 (3.0) | 0.246 | |
| Race | | | | 0.001 | |
| Asian (%) | 0 | 5 (3%) | 0 | | |
| Black/African American (%) | 9 (8%) | 3 (2%) | 0 | | |
| White (%) | 102 (92%) | 166 (95%) | 97 (100%) | | |
| Non-Hispanic (%) | 109 (98%) | 170 (98%) | 96 (99%) | 0.595 | |
| BMI (kg/m$^2$) | 26.9 (4.4) | 25.9 (3.9) | 25.4 (3.6) | 0.017 | |
| SBP (mmHg) | 133.8 (16.0) | 132.3 (15.5) | 134.1 (17.0) | 0.626 | |
| Having at least one _APOE_ ε4 allele (%) | 26 (23%) | 95 (55%) | 65 (67%) | <0.001 | |
| CDR-SB (SD) | 0.05 (0.24) | 1.69 (1.00) | 4.48 (1.88) | <0.001 | |
| CSF biomarkers | | | | | |
| Aβ42, mean (SD) in pg/mL | 209.8 (52.7) | 165.6 (52.4) | 142.5 (36.7) | <0.001 | |
| t-Tau, mean (SD) in pg/mL | 69.3 (30.2) | 105.2 (61.3) | 121.8 (57.7) | <0.001 | |
| p-Tau$_{181}$, mean (SD) in pg/mL | 27.0 (17.2) | 37.0 (21.3) | 43.2 (19.8) | <0.001 | |
| t-Tau/Aβ42 ≥ 0.39 (%) | 32 (29%) | 120 (69%) | 85 (88%) | <0.001 | |
| WU-sTREM2, mean (SD) in pg/mL | 2427 (774) | 2366 (726) | 2474 (802) | 0.611 | – |
| MSD-sTREM2, mean (SD) in pg/mL | 4692 (2274) | 4529 (2539) | 4347 (1975) | 0.628 | 0.871 |
| MSD-GRN, mean (SD) in pg/mL | 1598 (565) | 1612 (798) | 1626 (425) | 0.963 | 0.808 |
| TNF-α, mean (SD) in pg/mL | 1.91 (1.44) | 1.83 (1.20) | 1.73 (0.46) | 0.522 | 0.837 |
| sTNFR1, mean (SD) in pg/mL | 870 (227) | 904 (237) | 874 (249) | 0.434 | 0.353 |
| sTNFR2, mean (SD) in pg/mL | 1060 (512) | 1093 (319) | 1048 (284) | 0.597 | 0.299 |
| TGFβ1, mean (SD) in pg/mL | 109 (42) | 108 (42) | 107 (37) | 0.951 | 0.987 |
| TGFβ2, mean (SD) in pg/mL | 159 (39) | 161 (53) | 159 (47) | 0.398 | 0.168 |
| TGFβ3, mean (SD) in pg/mL | 9.2 (23.3) | 11.1 (28.3) | 14.2 (31.1) | 0.438 | 0.675 |
| IP-10, mean (SD) in ng/mL[a] | 5.47 (1.78) | 5.08 (2.06) | 5.01 (1.95) | 0.253 | 0.117 |
| IL-6, mean (SD) in pg/mL | 4.78 (3.33) | 5.27 (5.78) | 5.03 (4.74) | 0.676 | 0.970 |
| IL-7, mean (SD) in pg/mL | 1.49 (2.75) | 1.16 (0.79) | 1.41 (1.26) | 0.238 | 0.338 |
| IL-9, mean (SD) in pg/mL | 3.70 (1.91) | 3.33 (1.46) | 3.45 (1.65) | 0.239 | 0.416 |
| IL-10, mean (SD) in pg/mL | 5.80 (2.74) | 7.97 (28.55) | 5.57 (2.60) | 0.532 | 0.431 |
| IL-12p40, mean (SD) in pg/mL | 1.11 (1.03) | 5.81 (48.01) | 1.17 (1.03) | 0.470 | 0.159 |
| IL-21, mean (SD) in pg/mL | 12.93 (14.71) | 11.78 (12.10) | 12.10 (12.46) | 0.766 | 0.820 |
| sICAM-1, mean (SD) in pg/mL | 355.4 (184.1) | 400.2 (215.6) | 368.7 (187.3) | 0.154 | 0.100 |
| sVCAM-1, mean (SD) in ng/mL[a] | 41.3 (21.0) | 44.7 (26.1) | 48.7 (67.4) | 0.413 | 0.468 |

[a]Note two analytes have concentrations in the range of ng/mL.

decline beyond worse longitudinal cognitive trajectory (as measured by ADNI-Mem-EF), we interpreted these findings to support the role of sTNFR1 score as a prognostic biomarker in those with MCI due to predicted ADNC.

Two other findings suggest better prognosis associated with higher sTNFR1 scores. First, MCI participants with low core AD score and high sTNFR1 score were even less likely to decline ($p < 0.001$ by diagnostic conversion and $p = 0.023$ by CDR-SB ≥ 4 vs high AD score and high sTNFR1 score, median time to conversion of >60 months for both) than participants with two high scores. What's more, only 3 (6%) of 52 MCI participants with low AD scores had low sTNFR1 scores (vs. 22% among those with high AD scores, $p = 0.005$). Cox proportional hazard analysis in all MCI participants with high core AD scores found high sTNFR1 scores to halve the risks of cognitive/functional decline whether assessed by diagnostic conversion (hazards ratio=0.541, 95% CI 0.314-0.933; $p = 0.027$) or CDR-SB (hazards ratio = 0.454, 95% CI 0.265–0.778, $p = 0.004$).

**CSF sTREM2 associated with cognitive decline in dementia stage of AD.** We next analyzed the relationship between cognitive decline and baseline CSF biomarker scores in those with AD dementia, also using ADNI-Mem-EF and CDR-SB as

longitudinal outcome. In this group with greater likelihood of predicted ADNC, core AD score was less correlated with rates of cognitive decline than in MCI. Instead, sTREM2 score was inversely associated with rates of decline (Supplementary Table 8). Using a similar stratification strategy as MCI resulted in too few people with low core AD or sTREM2 score. We therefore divided people with high core AD score according to the median sTREM2 score ($n = 41$ for at/above and $n = 42$ for below score of −0.0689), with a smaller third group having low core AD scores ($n = 14$). In keeping with outcomes from MLL analysis, lower sTREM2 score was associated with faster conversion to dementia among those with high core AD scores (median 24 vs. >36 months, $p = 0.001$, Fig. 2a). After adjusting for age, sex, _APOE_ ε4, and baseline CDR-SB, greater sTREM2 scores were associated with reduced risks for decline (H.R. = 0.412, 95% CI 0.193–0.878, $p = 0.022$).

Since sTNFR1 score did not translate into a prognostic marker for AD dementia, we examined levels of sTNFR1 among the three AD dementia groups. Whereas sTREM2 levels expectedly differed between those with high and low sTREM2 scores, levels of sTNFR1 and progranulin—another protein loading onto the same sTNFR1 PC—did not (Fig. 2b). Substituting p-Tau$_{181}$ and sTREM2 concentrations for PC scores modestly reproduced the profiles of decline (median 24 vs. 36 months, $p = 0.028$, Fig. 2c).

**Table 2 PCA of CSF AD and inflammatory proteins in ADNI and two replication cohorts B and C, with loading values of proteins consistently associated with two or more PCs shown.**

| | | ADNI | | | Cohort B | Cohort C |
|---|---|---|---|---|---|---|
| | | NC (n = 85) | MCI (n = 129) | AD (n = 68) | NC, MCI, AD (n = 68) | NC (n = 47) |
| PC1 | t-Tau[a] | 0.724 | 0.871 | 0.831 | 0.577 | 0.734 |
| | p-Tau$_{181}$[a] | 0.805 | 0.866 | 0.826 | 0.474 | 0.727 |
| | Aβ42[a] | −0.752 | −0.745 | −0.612 | −0.781 | −0.595 |
| | t-Tau/Aβ42[a] | 0.710 | 0.675 | 0.774 | 0.923 | 0.925 |
| PC2 | sTNFR1[a] | 0.869 | 0.903 | 0.884 | 0.895 | 0.851 |
| | sTNRF2[a] | 0.784 | 0.883 | 0.881 | 0.859 | 0.850 |
| | sVCAM1[a] | 0.833 | 0.869 | 0.813 | 0.833 | 0.825 |
| | TGFβ1[a] | 0.533 | 0.469 | 0.561 | N.D. | N.D. |
| | sICAM1[a] | 0.502 | 0.467 | 0.661 | | |
| | IP-10 | | | 0.484 | 0.602 | N.D. |
| PC3 | MSD-sTREM2[a] | 0.876 | 0.883 | 0.876 | 0.639[b] | 0.787[b] |
| | WU-sTREM2[a] | 0.830 | 0.882 | 0.819 | | |
| | Progranulin | 0.519 | | 0.640 | N.D. | N.D. |
| PC4 | IL-6[a] | 0.811 | 0.799 | 0.828 | N.D. | N.D. |
| | IL-10[a] | 0.691 | 0.662 | 0.713 | 0.922 | 0.812 |
| PC5 | TGFβ2[a] | 0.885 | 0.820 | 0.754 | N.D. | N.D. |
| | TGFβ1[a] | 0.598 | 0.716 | 0.752 | N.D. | N.D. |
| | TGFβ3 | | 0.538 | 0.705 | N.D. | N.D. |
| PC6 | IL-7[a] | 0.866 | 0.810 | 0.797 | 0.912 | 0.942 |
| | TNF-α[a] | 0.753 | 0.429 | 0.707 | 0.757 | |
| | IL-9 | | 0.601 | 0.647 | | |
| PC7 | IP-10 | 0.722 | 0.705 | | | N.D. |
| | IL12-p40 | 0.686 | 0.772 | | N.D. | N.D. |
| | IL-9 | 0.571 | | | 0.865 | |
| PC8 | IL-21 | 0.458 | 0.904 | | N.D. | N.D. |
| | sICAM1 | | −0.624 | | 0.934 | |

[a]Proteins found in the same PC across ADNI diagnostic groups. Levels of IL-6, IL-12p40, IL-21, progranulin, and TGFβ1/2/3 were not measured in either replication cohort, and IP-10 was not measured in Cohort C.
[b]Different sTREM2 assays used between ADNI and the replication cohorts.

**Table 3 Factors associated with rates of cognitive decline in ADNI MCI participants with ADNI-Mem-EF or CDR-SB as outcome, 0-60 months after CSF collection (see Tables S5 and S6 for comparison between models without and with sTNFR1 scores, with improvement in Akaike Information Criterion (AIC) of 11.7 and 8.2; significant factors highlighted in bold with p < 0.00625 used for CSF biomarkers to adjust for multiple comparisons).**

| Cognitive measure | ADNI-Mem-EF | | CDR-SB | |
|---|---|---|---|---|
| | B (95% CI) | P | B (95% CI) | P |
| Months | 0.028 (0.002, 0.054) | 0.033 | −0.158 (−0.272, −0.045) | 0.006 |
| Baseline Cognitive measure | 0.943 (0.911, 0.996) | <0.001 | 0.906 (0.799, 1.013) | <0.001 |
| Baseline Cognitive measure × Months | 0.004 (0, 0.009) | 0.070 | 0.012 (0, 0.024) | 0.059 |
| Female sex | 0.013 (−0.052, 0.078) | 0.695 | | |
| Female sex × Months | −0.006 (−0.012, 0.001) | 0.019 | | |
| Age | 0 (−0.004, 0.005) | 0.871 | 0.007 (−0.007, 0.022) | 0.339 |
| Age X Months | −0.0006 (−0.0009, −0.0003) | <0.001 | 0.003 (0.001, 0.004) | <0.001 |
| APOE ε4+ | 0.057 (0, 0.116) | 0.060 | | |
| AD score | −0.046 (−0.080, −0.011) | 0.009 | −0.060 (−0.163, 0.042) | 0.246 |
| AD score × Months | −0.008 (−0.011, −0.005) | <0.001 | 0.029 (0.017, 0.040) | <0.001 |
| sTNFR1 score | −0.010 (−0.042, 0.022) | 0.547 | −0.026 (−0.139, 0.086) | 0.642 |
| sTNFR1 score × Months | 0.005 (0.002, 0.008) | <0.001 | −0.020 (−0.033, −0.008) | 0.002 |
| AD score X sTNFR1 score | 0.007 (−0.026, 0.040) | 0.667 | | |
| AD score X sTNFR1 score × Months | 0.002 (0, 0.005) | 0.078 | | |

Therefore, we confirmed previous findings that greater sTREM2 levels associated with a more benign course of AD dementia.

**CSF IL6 associated with cognitive change in NC but do not influence MCI risks.** Only cognitive changes in ADNI-Mem-EF were analyzed in NC as most (70 out of 104) had global CDR of 0 at the last follow-up visit (median of 60 months). LMM found baseline cognitive performance, age, and IL6 score each associated with rates of cognitive change through quadratic relationships, while core AD score had a more straightforward linear

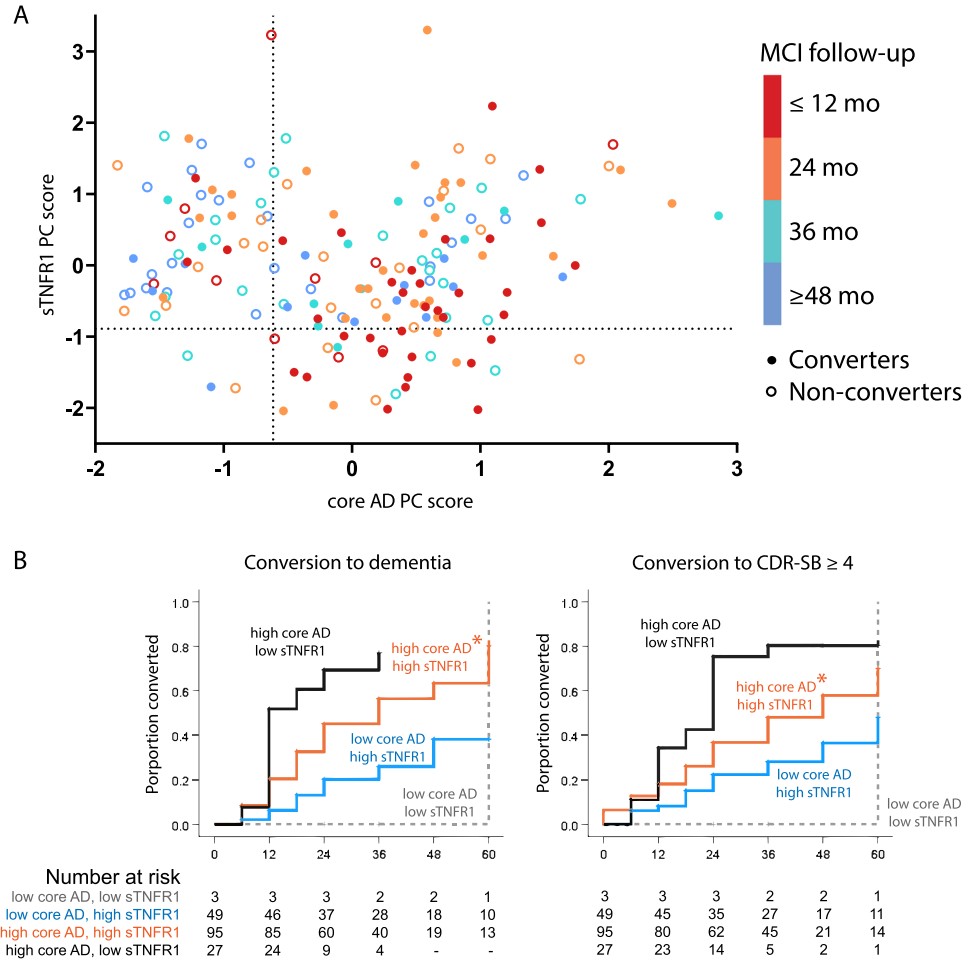

**Fig. 1 Relationship between two protein family scores (core AD and sTNFR1-related) and conversion from MCI to dementia.** Following principal component analysis, core AD principal component (PC) score corresponding to t-Tau/Aβ42 of 0.39 (predicted ADNC threshold) and sTNFR1 PC score corresponding to no net decline at this core AD score were used to stratify ADNI mild cognitive impairment (MCI) participants (**A**). Circle colors represent follow-up time, with converters shown as filled circles and non-converters shown as open circles. Participants with high core AD and low sTNFR1 scores had the shortest median time to conversion, followed by those with high scores in both and those with low core AD score (**B** *$p = 0.014$ with dementia diagnosis as outcome; †$p = 0.007$ with clinical dementia rating-sum of boxes score [CDR-SB] ≥ 4 as outcome). Source data are provided as a Source Data file.

relationship with cognitive decline (Supplementary Table 9). Stratifying NC participants according to core AD scores showed greater conversion to MCI ($p = 0.021$, Fig. 3a) or CDR 0.5 ($p = 0.023$, Fig. 3b) during the 60 months following CSF collection. Further division according to IL6 scores did not provide additional information on conversion risks most likely due to the overall low rates of decline.

**Validation of CSF sTNFR1 as a prognostic biomarker in MCI due to predicted ADNC.** Because individual protein levels are easier to translate as biomarkers in research and clinical settings than protein family scores, we also examined the relationship between the top prognostic biomarker scores for MCI and their constituent proteins to further validate in an independent cohort. Linear regression analysis in the ADNI MCI cohort mapped the core AD score of −0.614 onto z-transformed $\log_{10}$(p-Tau$_{181}$) value of 0.05 or p-Tau$_{181}$ level of 24.1 pg/mL. On the other hand, sTNFR1 score only modestly associated with sTNFR1 scores despite the high correlation. We therefore derived a regression-based prediction for sTNFR1 score using linear combinations of sTNFR1, sTNFR2, and sVCAM1 levels (all $\log_{10}$- and z-transformed). When

paired with p-Tau$_{181}$ levels, this predicted score ($y_{sTNFR1} = -0.101 + 0.633 \times z\log_{10}[sTNFR1] + 0.544 \times z\log_{10}[sVCAM1] - 0.230 \times z\log_{10}[sTNFR2]$) performed better in separating MCI participants into groups of different conversion rates than sTNFR1 levels alone (Fig. 4a).

We then applied the threshold values for p-Tau$_{181}$ and regression-predicted sTNFR1 score in an independent cohort of 49 MCI participants recruited and longitudinally characterized in Atlanta, including 33 participants from Cohort B in the cross-sectional PCA. Compared to the ADNI MCI participants, these MCI participants were younger (69.3 vs. 75.2, $p < 0.0001$), more likely to have self-reported race as Black (29% vs. 2%, $p < 0.0001$), and had lower CSF t-Tau and p-Tau$_{181}$ levels (Table 4). These MCI participants as well as their corresponding NC participants also had lower CSF sTNFR1 levels than ADNI participants with the same diagnosis (MCI: mean 645 vs. 904 pg/mL; NC: mean 587 vs. 836 pg/mL) which could result from pre-analytical CSF processing[8] (e.g., ADNI samples had one freeze-thaw cycle before aliquoting), but the NC-based z-transformation accounted for this systemic difference[8]. We confirmed that, compared to those with high p-Tau$_{181}$ levels and low predicted TNFR1 score, MCI participants with high p-Tau$_{181}$ and predicted sTNFR1 scores

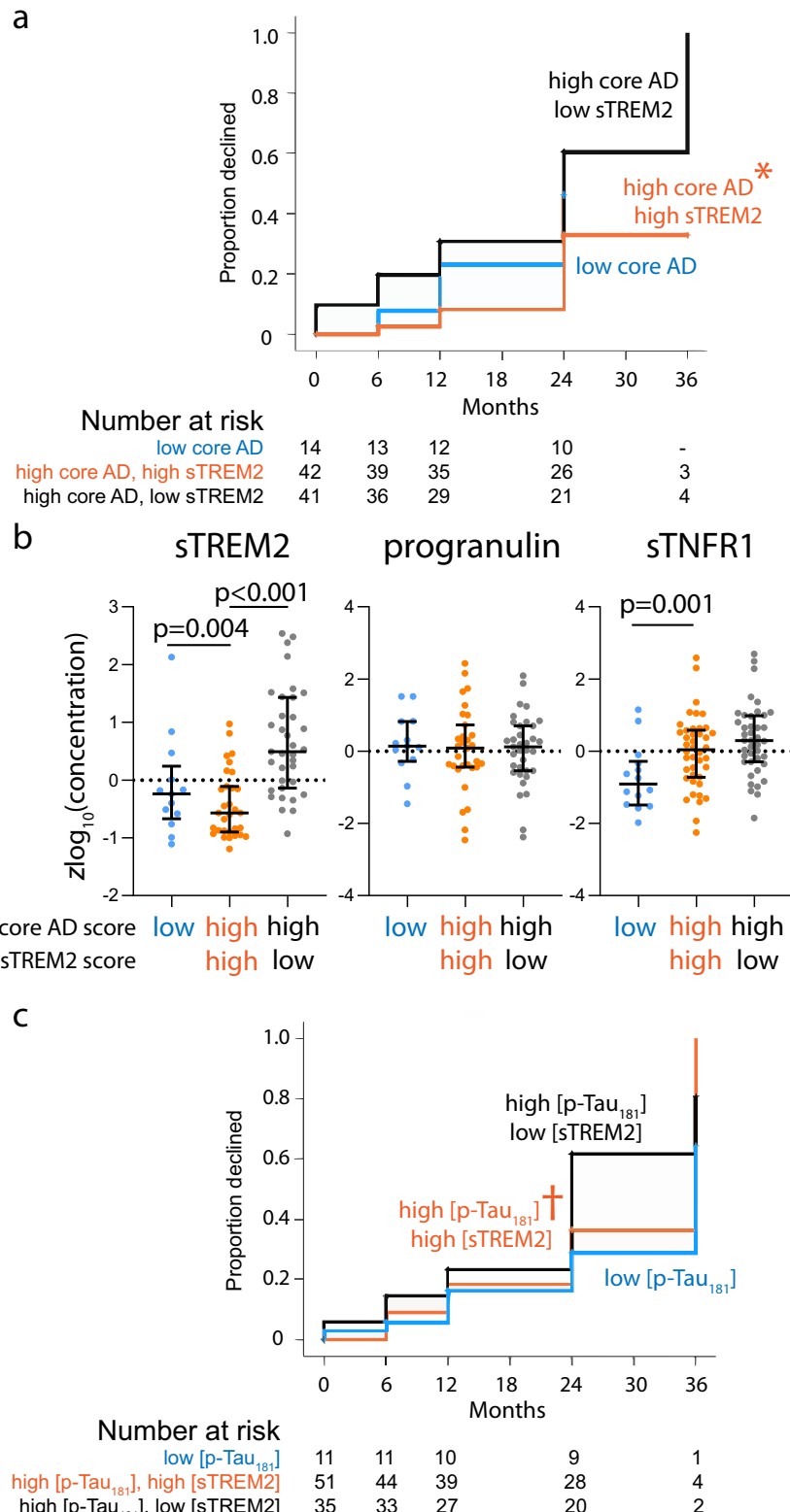

**Fig. 2 Relationship between two protein family scores (core AD and sTREM2) and cognitive decline in AD dementia.** AD dementia participants with high core AD and sTREM2 scores had slower conversion to a pre-set CDR-SB threshold (1.5 S.D. above the mean) than participants with low sTREM2 scores (**a** *$p = 0.001$). Concentrations of two proteins which did (sTREM2 and progranulin) and one did not (sTNFR1) load onto sTREM2 score showed greatest group-level differences in sTREM2 (**b** median and interquartile ranges shown). Substituting corresponding p-Tau181 and sTREM2 concentrations (shown in brackets to distinguish from **a**) for corresponding scores modestly reproduced the distinction (**c** †$p = 0.028$). Source data are provided as a Source Data file.

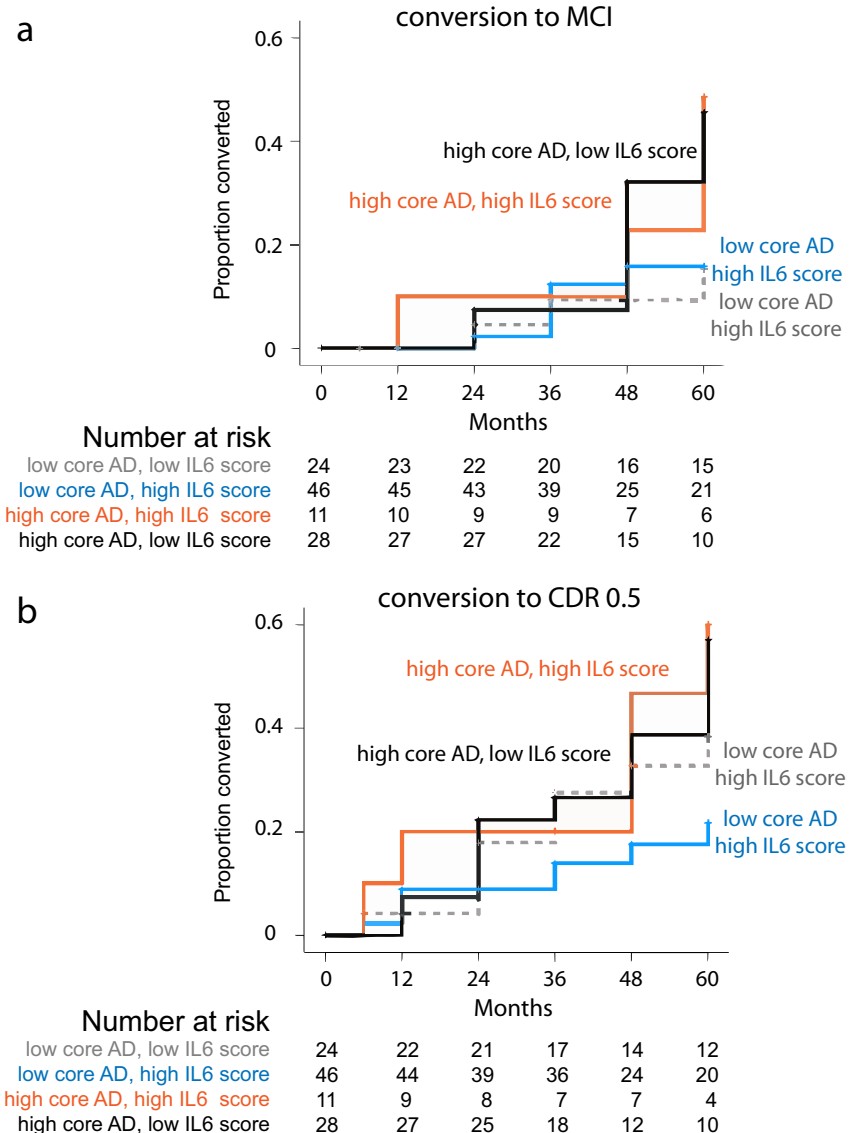

**Fig. 3 Relationship between two protein family scores (core AD and IL6) and cognitive decline in participants with normal cognition (NC).** NC participants with high core AD scores had earlier conversion to diagnosis of mild cognitive impairment (MCI; **a**, $p = 0.021$) or global clinical dementia rating score (CDR) of 0.5 (**b** $p = 0.023$), but further stratification according to IL6 score did not further refine long-term cognitive outcomes. Sample size in the NC cohort was 109 after excluding two participants with very rapid decline from NC to dementia. Source data are provided as a Source Data file.

($p = 0.038$) or low p-Tau$_{181}$ levels ($p = 0.068$) each had reduced likelihood of conversion to dementia (Fig. 4b).

## Discussion

Genetic and neuropathologic studies have pointed towards detrimental roles for inflammation in AD, but neuroinflammation can also trigger neuroprotective and pro-survival cascades[22,23]. We reproducibly identified four non-sTREM2 products of TACE/ADAM17 to co-vary across diagnosis in ADNI independent of core AD biomarkers, and a similar trend was replicated in two separate cohorts for three of these proteins which were measured. We further found higher levels of a PCA-derived score consisting of sTNFR1, sTNFR2, and sVCAM1—but not sTREM2—to associate with a halved risk of conversion among MCI participants with predicted ADNC in two independent cohorts. Consistent with the goal of a prognostic biomarker to identify likelihood of a clinical event or progression in patients who have the disease[19], we propose this

CSF-based prognostic biomarker can complement core AD diagnostic biomarkers in very early AD.

This study is no by means the first to systematically assess CSF inflammatory proteins in AD, but we used readily available commercial assays which we independently assessed for intermediate precision[24]. Others have examined CSF inflammatory proteins in AD, including sTNFR1 & sTNFR2[25,26] as well as sTREM2[12,27,28]. The large sample size in ADNI and moderate sample sizes in the two Atlanta cohorts allowed us to detect extraordinarily consistent PCs (families) across these cohorts, even when biomarkers within the same PC have been reported to derive from different cell types. Co-variance in their levels may afford the opportunity where one member of the family may be the best clinical biomarker while others better inform biological specificity. For example, sTNFR2 is more exclusively released by microglia and monocytes than sTNFR1 (which is released from all cell types), and non-endothelial VCAM1 is also expressed on microglia[29]. Therefore, sTNFR2/sVCAM1 may provide more cell-type specificity even though sTNFR1 had greater loading on the PC score.

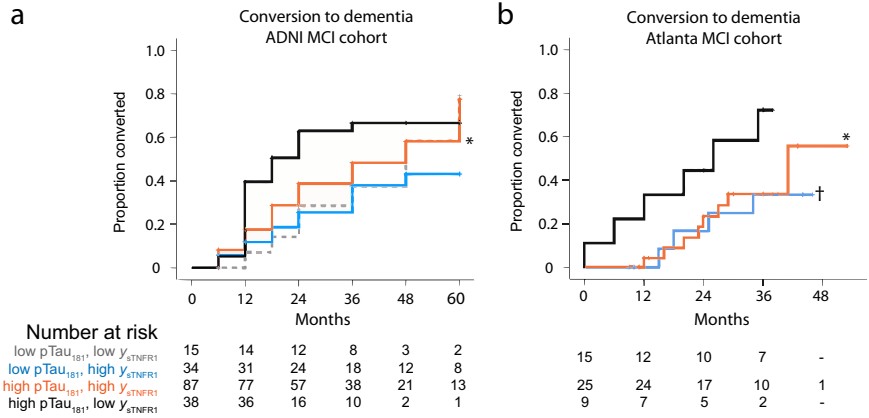

**Fig. 4 Mild cognitive impairment (MCI) conversion in the ADNI and an Atlanta-based MCI cohorts.** Conversion to dementia diagnosis based on the same p-Tau$_{181}$ levels and a regression-based prediction of sTNFR1 score ($y_{sTNFR1}$) reproduced the prognostic profiles from ADNI (**a**) in the replication cohort (**b** $n = 49$). *Lower conversion compared to those with p-Tau$_{181} \geq 24.1$ pg/mL but low $y_{sTNFR1}$ ($p = 0.049$ in ADNI and $p = 0.038$ in the Atlanta cohort). †Two subgroups with p-Tau$_{181} < 24.1$ were combined due to small numbers, $p = 0.068$ vs. high p-Tau$_{181}$ and low $y_{sTNFR1}$. Source data are provided as a Source Data file.

| Table 4 MCI participants in replication cohort ($n = 49$). | | |
|---|---|---|
| | **Replication MCI cohort** | ***p* (vs. ADNI MCI)** |
| Male (%) | 26 (65%) | 0.183 |
| Age, mean (SD) | 69.3 (7.9) | <0.0001 |
| Education, mean (SD) | 16.5 (2.5) | 0.126 |
| Race | | <0.0001 |
| Asian (%) | 0 | |
| Black/African American (%) | 14 (29%) | |
| White (%) | 35 (71%) | |
| Non-Hispanic (%) | 49 (100%) | 0.578 |
| Having at least one *APOE* ε4 allele (%) | 14/28 (50%) | 0.684 |
| CSF biomarkers | | |
| Aβ42, mean (SD) in pg/mL | 179.7 (142.9) | 0.451 |
| t-Tau, mean (SD) in pg/mL | 62.5 (36.6) | <0.0001[a] |
| p-Tau$_{181}$, mean (SD) in pg/mL | 29.6 (17.3) | 0.0001[a] |
| t-Tau/Aβ42 ≥ 0.39 (%) | 27 (55%) | 0.088 |
| sTNFR1 | 645.5 (179.4) | <0.0001[a] |

[a]Log10-transformed values were analyzed.

Conversely, while some TACE/ADAM17 sheddase products were reliably found in the same PC, others—TNFα, sTREM2—were not. This is not surprising for biological and statistical reasons. Biologically, factors such as circulation, interstitial exchange[30], bulk flow[31,32], and lymphatic clearance[33] can all influence measured CSF protein levels beyond surface cleavage. Statistically, PCA exploits underlying proteins levels' data structures to reduce the number of dimensions while maximizing variance explained, but a modest correlation between two variables across hidden clusters may mask the variables' more independent relationships[34]. In other words, we were more interested in associated protein level changes independent of AD status/biomarkers than proteins which differed in levels between NC and AD dementia. Our data-driven approach therefore provides information not otherwise available through knowledge-based pathway analysis, but these loading profiles should still be prospectively tested using drugs known to alter one or more of the proteins.

CSF sTREM2 levels were previously investigated in ADNI due to the genetic linkage between rare *TREM2* mutations and AD

risks[12,35]. We do not dispute the finding that, in a sufficiently large cohort, elevated CSF sTREM2 can provide prognostic information in MCI through group-level correlation with sTNFR1-related proteins. Because measured CSF proteins are influenced by biological and technical factors, it is likely premature to advance sTNFR1- or sTREM2-elevating therapies in AD based entirely on their association with rates of cognitive change. Whether the stage-specific nature of these prognostic biomarkers (sTNFR1/sTNFR2/sVCAM-1 for MCI stage of AD, sTREM2 for dementia stage of AD) also needs to be further investigated for their relationships to biological processes grouped under the umbrella term of neurodegeneration. Since microglia—the presumed cell of origin for these proteins—can display different functional phenotypes according to the local environment, poorer prognosis associated with lower sTNFR1 and sTREM2 at different AD stage may begin to shape a framework for the temporal sequencing of microglia dysfunctions.

Our study has a number of limitations, including the selective nature of the ADNI cohort, the smaller sample size of the Atlanta MCI cohort, not correlating with measures of brain atrophy or ischemia on MRI, and not having adequate power in longitudinal analysis to examine the effect of race on biomarker-based prognosis. Nevertheless, we show in two well-characterized independent cohorts that higher levels of CSF sTNFR1-related proteins (sTNFR1, sTNFR2, sVCAM1) are associated with reduced cognitive decline in those with MCI independent of markers of ADNC. Our studies developed a reproducible scheme to nominate and analyze CSF inflammatory as biomarkers in neurodegeneration, and results of such methods validated prior findings from smaller studies[36,37]. The inflammatory alterations we identified potentially reflect microglial phenotype evolution along the AD disease continuum, and autopsy- or PET-based analysis of people in early AD stages is necessary to confirm this. Because we demonstrated here the high intermediate precision of commercial assays for sTNFR1, sTNFR2, and sVCAM-1, these biomarkers can be readily introduced into existing workflows to provide prognostic information and improve clinical trial design.

## Methods

**Ethics approval.** This study was approved by the Emory University Institutional Review Board, the National Institutes on Aging, and the ADNI Resource Allocation and Review Committee. The ADNI was launched in 2003 as a public-private partnership, led by Principal Investigator Michael W. Weiner, MD. The primary goal of ADNI has been to test whether serial magnetic resonance imaging, positron emission tomography, other biological markers, and clinical and

neuropsychological assessment can be combined to measure the progression of MCI and early AD dementia. All participants or legally-authorized representatives provided written informed consent to participate in the studies.

**Study design and participants.** A cross-sectional study design was used for PCA of CSF AD and inflammatory proteins. ADNI participants (NCT00106899) were selected to have adequate representation of healthy control participants with normal cognition (NC; $n = 111$) to establish normative range of inflammatory proteins; sufficient numbers of MCI ($n = 174$) and AD dementia ($n = 97$) to identify CSF proteins associated with rates of longitudinal cognitive and functional decline; maximize overlap with participants with measured levels of complement 3 and complement factor H[7]; and match sample availability. A power calculation was performed to arrive at a total sample size of at least 250 (MCI and AD dementia together) to have a power of 0.95 to detect an effect size of 0.15 for fixed factors (sex, APOE ε4 status) and covariates (age, baseline cognitive function, two inflammatory protein levels, two core AD biomarker levels) in LMM of longitudinal cognitive decline with $p = 0.05$ if the repeated measure correlation is 0.3, and has power of 0.83 to detect an effect size of 0.15 with $p = 0.02$ if the repeated measure correlation is 0.2.

Other than ADNI participants, two other previously published cohorts were included for replication of the relationship among CSF biomarkers (Supplementary Methods; Table S5). Cohort B (NCT02089555; PI: WTH) was a cohort of older white ($n = 68$) and Black American ($n = 62$) participants with NC, MCI, and AD dementia recruited from Georgia who underwent detailed prospective baseline neurological, neuropsychological, CSF, and MRI characterization for identifying race-associated biomarker differences[38]. Participants in this study were recruited from the Emory Cognitive Neurology Clinic, Emory Alzheimer's Disease Research Center, or community events in the greater Atlanta area. Cohort C (NCT00135226; PI: WW) was a cohort of middle-aged to older white ($n = 47$) and Black participants with NC ($n = 21$) who underwent detailed prospective baseline and longitudinal neuropsychological, CSF, and MRI characterization for identifying effect of race on vascular and AD biomarkers[9,39]. Participants in this study were recruited from the Emory Alzheimer's Disease Research Center, community events, or dementia caregiving groups in the greater Atlanta. Because of race-based differences in CSF tau-related and inflammatory protein levels[9,38,40], only white participants from Cohorts B & C ($n = 115$) were used to replicate PCA findings from ADNI.

To validate the relationship between p-Tau$_{181}$-sTNFR1 combination and longitudinal decline in MCI, an MCI validation cohort ($n = 49$) was created by combining white and Black MCI participants from Cohort B ($n = 33$) who underwent longitudinal follow-up and a separate group of older white and Black MCI participants ($n = 16$) who underwent similar longitudinal clinical and neuropsychological evaluations following detailed baseline neurological, neuropsychological, CSF, and MRI analysis.

**CSF biomarker measurements.** For ADNI, blinded CSF samples were shipped from the Biomarker Core (University of Pennsylvania) to Emory University for analysis in 2018. CSF analysis was performed by two skilled research scientists experienced in multiplex assays blinded to diagnosis and other subject-level information. All samples were run in duplicate with six CSF standards on each plate, and CSF inflammatory protein levels were normalized across plates using the six CSF standard values. ADNI CSF samples were first randomized across twelve 96-well plates, and each batch was analyzed for levels of all 15 proteins during the same two-day block to avoid freeze-thawing. Assay 1 included sTNFR1 and sTNFR2; assay 2 included TGFβ 1,2, and 3; assay 3 included IL-21; assay 4 included sICAM-1 and sVCAM-1; and assay 5 included TNFα, IL-6, IL-7, and IL-10 for the first 98 participants, and TNFα, IL-6, IL-7, IL-10, IL-9, IL-12p40, and IP-10 for the remaining 280 participants (see Missing Data). All samples were run in duplicate with six CSF standards on each plate. CSF inflammatory protein levels were normalized across plates using the six CSF standard values, and intermediate precision for each analyte was then calculated using inter-plate coefficient of variation: 9.38% for TNFα, 2.85% for sTNFR1, 3.09% for sTNFR2, 10.99% for sVCAM1, 9.86% for sICAM1, 6.30% for IL-6, 14.68% for IL-7, 9.24% for IL-9, 16.51% for IL-10, 6.41% for IL-12p40, 20.6% for IL-21, 4.83% for IP-10, 8.62% for TGFβ1, 7.62% for TGFβ2, 7.70% for TGFβ3. Values for Aβ42, t-Tau, p-Tau$_{181}$, sTREM2 (from MSD and Washington University [WU])[12], and soluble progranulin levels were obtained from ADNI. CSF biomarkers for the three Atlanta cohorts were analyzed by the same laboratory research scientists using the same volume, CSF standards, blinding, and randomization scheme[9,38].

**Missing data.** Among 382 ADNI participants whose CSF samples were available for this study, the first 100 samples (26%) had 12 proteins measured (all except IL-9, IL-12p40, and IP-10). The remaining 280 samples (74%) had all 15 CSF inflammatory biomarker measured. The 100 samples were from participants randomized across the three diagnostic categories (25 NC, 53 MCI, 22 AD dementia), and they were similar to the remaining participants according to age, sex, APOE ε4 status, and predicted ADNC within each diagnostic category.

319 of the 382 (83%) previously had sTREM2 and progranulin levels measured, with a core group of 241 participants (75 NC, 104 MCI, 62 AD dementia) having

all CSF proteins (AD biomarkers, sTREM2, progranulin, and inflammatory proteins reported here) measured. PCA was first performed by excluding participants with at least one missing value (listwise), and then with missing values imputation by means and expectation maximization (see Statistical analysis below).

**Statistical analysis.** Data used in the preparation of this article—other than levels of 15 CSF inflammatory proteins—were obtained from the ADNI database (adni.loni.usc.edu). All analyses were performed in IBM SPSS 26.0 (Armonk, NY). Two sided tests were used for all analyses. For CSF inflammatory proteins, outliers were determined as log$_{10}$- and z-transformed values greater than 4 or less than −4. Because we failed to detect individuals who had consistent outlier protein levels, no outlier inflammatory protein levels were excluded. However, two NC participants had atypical profile of functional decline (Fig. S1) and were excluded from LMM analysis. Two-tailed tests were used in all statistical analysis. CSF inflammatory protein levels were assessed for their normality using Kolmogorov-Smirnov test. All markers except MSD-sTREM2 showed non-normal distribution and were log$_{10}$ transformed (as were t-Tau and p-Tau$_{181}$ for their skewed distribution). Because mean inflammatory protein levels ranged from 1.10 pg/mL (IL12-p40) to 41.29 ng/mL (sVCAM1), we chose to z-transform their levels for better assessment of their association with rates of decline relative to each other according to mean and standard deviation (S.D.) values for NC participants at baseline (Supplementary Table 2).

Analysis of co-variance (ANCOVA) was first used to determine inflammatory protein differences across diagnostic categories and predicted ADNC status, adjusting for age, sex, and APOE ε4 status (Supplementary Table 3). All participants with available measures were included in the ANCOVA for each protein. To identify orthogonal eigenvectors to reduce the number of protein variables and align correlated measures with shared variance, PCA with varimax rotation was then used to separate proteins into PCs using eigenvalues >0.7 as recommended by Jolliffe[41]. This was first performed in participants with all protein values available ($n = 241$) and then with missing values replaced with means (values for MCI participants are shown in Supplementary Table 4). Factor loading ≥0.400 in both missing data handling methods were considered consistent elements of each PC for inclusion in Table 2.

Because excluding cases or replacing with means can bias the outcomes[42], we additionally confirmed PCA results with imputation through expectation maximization to reduce bias[43–45]. In the ADNI MCI cohort, this produced identical membership (loading ≥ 0.400) for PC1, PC2, PC3, PC4, PC5, PC6, and PC8, with PC7 now having additional loading by IL-10, TNFα and TGFβ3. Thus, missing data handling through three separate approaches all generated highly reproducible PCs.

LMM was used to identify potential predictors of future cognitive decline. Because PCA consistently placed two of the three proteins not measured in all participants (IP-10, IL12-p40) in the same PC, LMM first only focused on participants with measured IP-10 and IL12-p40 levels, but then all participants if IP-10/IL-12 score did not associate with rates of cognitive or functional decline at $p < 0.10$. Cognitive decline in ADNI was analyzed using validated composite Memory (ADNI-Mem) and Executive Function (ADNI-EF) scores previously generated from subtests targeting respective functions through item-response theory and bi-factor confirmatory factor analysis to optimize these scores for longitudinal tracking[46,47]. ADNI-Mem subtests included Rey Auditory Visual Learning Test (learning & recall), AD Assessment Scale—Cognitive Subscale, word recall from Mini-Mental State Examination, and Logical Memory I and II from the Wechsler Memory-Test Revised. ADNI-EF subtests included category fluency, oral Trail Making Test A & B, Digit Span Backwards, Digit Symbol Substitution Test from the Wechsler Adult Intelligence Scale Revised, and Clock Drawing Test. Each factor was normalized to have a mean of 0 and standard deviation of 1, equivalent to a z-score transformation. Because CDR-SB is used to track longitudinal cognitive performance[48] but was not included in ADNI-Mem or ADNI-EF, cognitive decline in ADNI was also analyzed according to CDR-SB.

In LMM, time (in months), Time[2], and Time[3] were included as fixed and random variables to determine if linear, quadratic, and cubic models best described longitudinal cognitive decline. Akaike Information Criterion (AIC) used to assess whether a model incorporating additional factors (e.g., Time[2], core AD score, and interaction terms) was better than a simpler model without overfitting. This creates the baseline models without additional PCs. Models incorporating Time only achieved substantially better AIC (Δ»10) than models additionally incorporating Time[2] and Time[3] terms for MCI and AD dementia, but higher order terms improved the fit for long-term cognitive decline in NC. Subsequent models incorporating Time, demographic variables, and biomarker PC scores were assessed using criteria of substantial (ΔAIC > 10), moderate (7 ≥ ΔAIC ≥ 4), or minimal (2 ≥ ΔAIC) improvement. For each diagnostic category, ADNI-Mem-EF and CDR-SB were each used as the time-dependent variable of cognitive outcome. PC scores were tested in a stepwise fashion to determine each score's impact on AIC.

To illustrate prognostic impact of CSF biomarker PC scores, KM survival analysis was used to determine their relationship to cognitive decline. Significant cognitive decline in MCI was assessed by diagnostic conversion (to dementia) as well as CDR-SB. Based on prior data from 792 participants longitudinally followed at the Knight AD Research Center at Washington University, 1 standard deviation

above the mean for MCI and 1 standard deviation below the mean for AD dementia both had CDR-SB of 3.8[48]. Therefore, CDR-SB ≥ 4 was selected for MCI participants for significant worsening. Cox-proportional hazard analysis was further used among ADNI MCI participants to determine impact of CSF biomarker PC score on conversion, adjusting for age, sex, and APOE genotype. Core AD biomarker and sTNFR1 scores were initially used in these analyses, but we also derived simplified measures more suitable for clinical application. Linear regression analysis was first used to determine the p-Tau$_{181}$ concentration (in pg/mL) corresponding to the threshold core AD biomarker. Linear regression was also used to determine sTNFR1 concentration corresponding to the threshold sTNFR1 score. Because this score only provided prognostic information in conversion according to CDR-SB but not consensus diagnosis, we further calculated a predict sTNFR1 score ($y_{sTNFR1}$) using linear recombination of sTNFR1, sTNFR2, and sVCAM-1 concentrations. Values for p-Tau$_{181}$ and $y_{sTNFR1}$ were then tested in the Atlanta replication cohort for diagnostic conversion (CDR-SB not available for many participants).

A similar conversion analysis was used to assess biomarker scores' effects on significant decline in AD dementia in the 36 months following CSF collection, with a threshold of CDR-SB > 7.8 (1.5 standard deviation above the mean) derived from the Knight cohort[48]. For NC, conversion to MCI or global CDR ≥ 0.5 was examined in the 60 months following CSF collection.

**Role of the funding source.** The funding sources had no role in the study design; in collection, analysis, and interpretation of data; in the writing of the report; and in decision to submit the paper for publication.

**Reporting summary.** Further information on research design is available in the Nature Research Reporting Summary linked to this article.

## Data availability
All ADNI data (including CSF inflammatory protein measures) are available for public access at adni.loni.usc.edu contingent on adherence to the ADNI Data Use Agreement, and all data from the Atlnta cohorts are available from RUResearch Data Portal (https://rucore.libraries.rutgers.edu/research/) and available as a Source File. Source data are provided with this paper.

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

## Acknowledgements

Funding: This study is funded by NIH R01 AG 054046, R21 AG 043885, K01AG042498, and Bobbie Bailey Foundation (Atlanta, GA). Data collection and sharing for ADNI is funded by NIH U01 AG024904 and DOD W81XWH-12-2-0012. ADNI is funded by the National Institute on Aging, the National Institute of Biomedical Imaging and Bioengineering, and through generous contributions from the following: AbbVie, Alzheimer's Association; Alzheimer's Drug Discovery Foundation; Araclon Biotech; BioClinica, Inc.; Biogen; Bristol-Myers Squibb Company; CereSpir, Inc.; Cogstate; Eisai Inc.; Elan Pharmaceuticals, Inc.; Eli Lilly and Company; EuroImmun; F. Hoffmann-La Roche Ltd and its affiliated company Genentech, Inc.; Fujirebio; GE Healthcare; IXICO Ltd.; Janssen Alzheimer Immunotherapy Research & Development, LLC.; Johnson & Johnson Pharmaceutical Research & Development LLC.; Lumosity; Lundbeck; Merck & Co., Inc.; Meso Scale Diagnostics, LLC.; NeuroRx Research; Neurotrack Technologies; Novartis Pharmaceuticals Corporation; Pfizer Inc.; Piramal Imaging; Servier; Takeda Pharmaceutical Company; and Transition Therapeutics. The Canadian Institutes of Health Research is providing funds to support ADNI clinical sites in Canada. Private sector contributions are facilitated by the Foundation for the National Institutes of Health (www.fnih.org). The grantee organization is the Northern California Institute for Research and Education, and the study is coordinated by the Alzheimer's Therapeutic Research Institute at the University of Southern California. ADNI data are disseminated by the Laboratory for Neuro Imaging at the University of Southern California.

## Author contributions

W.T.H. and W.W. were responsible for conception/design the study; W.T.H., T.O., W.W., A.K., and J.C.H. were responsible for acquisition, analysis, and interpretation of data; W.T.H. and J.C.H. were responsible for drafting the work, and T.O., W.W., and A.K. were responsible for critical revision of the work for important intellectual content. All authors have given final approval for the version submitted, and agree to be accountable for all aspects of the work.

## Competing interests

W.T.H. has served as a consultant to ViveBio LLC, Biogen Inc., Fujirebio Diagnostics, Apellis Pharmaceuticals, and AARP Inc.; received research support from Fujirebio USA; and has a patent on CSF-based diagnosis of FTLD-TDP (assigned to Emory University). All other authors declare no competing interests.

## Additional information

## Alzheimer's Disease Neuroimaging Initiative

Michael Weiner[3], Paul Aisen[4], Ronald Petersen[5], Clifford R. Jack Jr.[5], William Jagust[6], John Q. Trojanowski[7], Arthur W. Toga[4], Laurel Beckett[8], Robert C. Green[9], Andrew J. Saykin[10], John Morris[11], Richard J. Perrin[11], Leslie M. Shaw[7], Zaven Kachaturian[12], Maria Carrillo[13], William Potter[14], Lisa Barnes[15], Marie Bernard[16], Hector González[17], Carole Ho[18], John K. Hsiao[16], Eliezer Masliah[16], Donna Masterman[19], Ozioma Okonkwo[20], Laurie Ryan[16], Nina Silverberg[16], Adam Fleisher[21], Tom Montine[22], Jeffrey A. Kaye[23], Lisa C. Silbert[23], Lon S. Schneider[4], Sonia Pawluczyk[4], Mauricio Becerra[4], James Brewer[17], Judith L. Heidebrink[24], David Knopman[5], Javier Villanueva-Meyer[25], Rachelle S. Doody[25], Joseph S. Kass[25], Yaakov Stern[26], Lawrence S. Honig[26], Akiva Mintz[26], Beau Ances[11], Mark A. Mintun[11], David Geldmacher[27], Marissa Natelson Love[27], Hillel Grossman[28], Martin A. Goldstein[28], Raj C. Shah[15], Melissa Lamar[15], Ranjan Duara[29], Maria T. Greig-Custo[29], Marilyn Albert[30], Chiadi Onyike[30], Amanda Smith[31], Martin Sadowski[32], Thomas Wisniewski[32], Melanie Shulman[32], P. Murali Doraiswamy[33], Jeffrey R. Petrella[33], Olga James[33], Jason H. Karlawish[7], David A. Wolk[7], Charles D. Smith[34], Gregory A. Jicha[34], Riham El Khouli[34], Oscar L. Lopez[35], Anton P. Porsteinsson[36], Gaby Thai[37], Aimee Pierce[37], Brendan Kelley[38], Trung Nguyen[38], Kyle Womack[38], Allan I. Levey[1,39], James J. Lah[1,39], Jeffrey M. Burns[40], Russell H. Swerdlow[40],

William M. Brooks[40], Daniel H. S. Silverman[41], Sarah Kremen[41], Neill R. Graff-Radford[42], Martin R. Farlow[10], Christopher H. van Dyck[43], Adam P. Mecca[43], Howard Chertkow[44], Susan Vaitekunis[44], Sandra Black[45], Bojana Stefanovic[45], Chinthaka Heyn[45], Ging-Yuek Robin Hsiung[46], Vesna Sossi[46], Elizabeth Finger[47], Stephen Pasternak[47], Irina Rachinsky[47], Ian Grant[48], Emily Rogalski[48], M.-Marsel Mesulam[48], Nunzio Pomara[49], Raymundo Hernando[49], Antero Sarrael[49], Howard J. Rosen[3], Bruce L. Miller[3], David Perry[3], Raymond Scott Turner[50], Reisa A. Sperling[9], Keith A. Johnson[9], Gad A. Marshall[9], Jerome Yesavage[22], Joy L. Taylor[22], Steven Chao[22], Christine M. Belden[51], Alireza Atri[51], Bryan M. Spann[51], Ronald Killiany[52], Robert Stern[52], Jesse Mez[52], Thomas O. Obisesan[53], Oyonumo E. Ntekim[53], Alan Lerner[54], Paula Ogrocki[54], Curtis Tatsuoka[54], Evan Fletcher[8], Pauline Maillard[8], John Olichney[8], Charles DeCarli[8], Vernice Bates[55], Horacio Capote[55], Michael Borrie[56], T.-Y. Lee[56], Rob Bartha[56], Sterling Johnson[20], Sanjay Asthana[20], Cynthia M. Carlsson[20], Allison Perrin[57], Douglas W. Scharre[58], Maria Kataki[58], Rawan Tarawneh[58], David Hart[59], Earl A. Zimmerman[59], Dzintra Celmins[59], Delwyn D. Miller[60], Hristina Koleva[60], Hyungsub Shim[60], Jeff D. Williamson[61], Suzanne Craft[61], Jo Cleveland[61], Brian R. Ott[62], Jonathan Drake[62], Geoffrey Tremont[62], Marwan Sabbagh[63], Aaron Ritter[63], Jacobo Mintzer[64], Joseph Masdeu[65], Jiong Shi[66], Paul Newhouse[67], Steven Potkin[68], Stephen Salloway[69], Paul Malloy[69], Stephen Correia[69], Smita Kittur[70], Godfrey D. Pearlson[71], Karen Blank[71], Laura A. Flashman[72], Marc Seltzer[72], Athena Lee[73], Norman Relkin[73] & Gloria Chiang[73]

[3]UC San Francisco, San Francisco, CA, USA. [4]USC, Los Angeles, CA, USA. [5]Mayo Clinic, Rochester, MN, USA. [6]UC Berkeley, Berkeley, CA, USA. [7]U Pennsylvania, Philadelphia, PA, USA. [8]UC Davis, Davis, CA, USA. [9]Brigham and Women's Hospital, Harvard Medical School, Cambridge, MA, USA. [10]Indiana University, Indianapolis, IN, USA. [11]Washington University St. Louis, St. Louis, MO, USA. [12]Prevent Alzheimer's Disease 2020, Rockville, MD, USA. [13]Alzheimer's Association, Chicago, IL, USA. [14]National Institute of Mental Health, Bethesda, MD, USA. [15]Rush University, Chicago, IL, USA. [16]National Institute on Aging, Bethesda, MD, USA. [17]University of California, San Diego, CA, USA. [18]Denali Therapeutics, San Francisco, CA, USA. [19]Biogen, Cambridge, MA, USA. [20]University of Wisconsin, Madison, WI, USA. [21]Eli Lilly, Indianapolis, IN, USA. [22]Stanford University, Palo Alto, CA, USA. [23]Oregon Health & Science University, Portland, OR, USA. [24]University of Michigan, Ann Arbor, MI, USA. [25]Baylor College of Medicine, Houston, TX, USA. [26]Columbia University Medical Center, New York, NY, USA. [27]University of Alabama, Birmingham, AL, USA. [28]Mount Sinai School of Medicine, New York, NY, USA. [29]Wein Center, Miami Beach, FL, USA. [30]Johns Hopkins University, Baltimore, MD, USA. [31]University of South Florida: USF Health Byrd Alzheimer's Institute, Tampa, FL, USA. [32]New York University, New York, NY, USA. [33]Duke University, Durham, NC, USA. [34]University of Kentucky, Lexington, KY, USA. [35]University of Pittsburg, Pittsburg, PA, USA. [36]University of Rochester, Rochester, NY, USA. [37]UC Irvine, Irvine, CA, USA. [38]University of Texas Southwestern Medical School, Dallas, TX, USA. [39]Emory Goizueta Alzheimer's Disease Research Center, Atlanta, GA, USA. [40]University of Kansas Medical Center, Kansas City, USA. [41]UC Los Angeles, Los Angeles, CA, USA. [42]Mayo Clinic, Jacksonville, FL, USA. [43]Yale University School of Medicine, New Haven, CT, USA. [44]McGill University, Montreal, QC, Canada. [45]Sunnybrook Health Sciences, Toronto, ON, Canada. [46]University of British Columbia, Vancouver, BC, Canada. [47]St. Joseph's Health Care, London, ON, Canada. [48]Northwestern University, Chicago, IL, USA. [49]Nathan Kline Institute, New York, NY, USA. [50]Georgetown University Medical Center, Washington, DC, USA. [51]Banner Sun Health Research Institute, Phoenix, AZ, USA. [52]Boston University, Boston, MA, USA. [53]Howard University, Washington, DC, USA. [54]Case Western Reserve University, Cleveland, OH, USA. [55]Dent Neurologic Institute, New York, NY, USA. [56]Parkwood Institute, London, ON, Canada. [57]Banner Alzheimer's Institute, Phoenix, AZ, USA. [58]Ohio State University, Columbus, OH, USA. [59]Albany Medical College, Albany, NY, USA. [60]University of Iowa College of Medicine, Iowa City, IA, USA. [61]Wake Forest University Health Sciences, Winston Salem, NC, USA. [62]Rhode Island Hospital, Providence, RI, USA. [63]Cleveland Clinic Lou Ruvo Center for Brain Health, Las Vegas, NV, USA. [64]Roper St. Francis Healthcare, Charleston, SC, USA. [65]Houston Methodist Neurological Institute, Houston, TX, USA. [66]Barrow Neurological Institute, Phoenix, AZ, USA. [67]Vanderbilt University Medical Center, Nashville, TN, USA. [68]Long Beach VA Neuropsychiatric Research Program, Long Beach, CA, USA. [69]Butler Hospital Memory and Aging Program, Providence, RI, USA. [70]Neurological Care of CNY, Syracuse, NY, USA. [71]Hartford Hospital—Olin Neuropsychiatry Research Center, Hartford, CT, USA. [72]Dartmouth-Hitchcock Medical Center, Lebanon, NH, USA. [73]Cornell University, New York, NY, USA.

