## [Peer Review File · Nature Communications]

REVIEWER COMMENTS

Reviewer #1 (Remarks to the Author):

There are substantial genetic and neuropathologic studies investigating the detrimental role of inflammation in AD, but they reported inconsistent associations of CSF inflammatory proteins with Alzheimer's core biomarkers and cognitive decline. The authors reproducibly identified four non-sTREM2 products of TACE/ADAM17 to co-vary across diagnosis, independent of core AD biomarkers. They also found increased level of CSF sTNFR1-related proteins – but not sTREM2 – to halve the risks of 36-month decline among MCI subjects with predicted Alzheimer's pathology. This research has certain scientific significance. But there remains some deficiencies. Recommendations are below:

- 1) The method is not well arranged. In the Ethics Approval section, authors describe: "ADNI subjects were selected to have adequate representation of healthy control subjects with normal cognition (NC; n=111) to establish normative range of inflammatory proteins ..." which should be in the Study design and participants section.
- 2) In the second paragraph of the method, authors describe: "The objective of this study is to validate previous in-laboratory findings that CSF inflammatory proteins form reproducible PCs, and ..." The term PC should be given the complete spelling for it occurs at the first time. Besides, most of the content in this paragraph belongs to statistical methods rather than study design.
- 3) In the third paragraph of the method, authors mentioned other cohorts independent of ADNI. These cohorts should be fully described, not merely cited. It leads to confusion whether these cohorts were from large public database or collected by the authors' team.
- 4) To validate the sTNFR1 as a novel prognostic biomarker for MCI, authors chose subsets from the first and third cohorts which had rather small sample sizes (n = 33 and 16, respectively). Did the authors calculate the power of test?
- 5) Methods of the cognition assessment should be described in detail, especially the cognitive z-score.
- 6) The inclusion and exclusion criteria of participants enrolled in the study should be stated clearly in the method section.
- 7) Authors should better describe the prognostic outcome indicators for different diagnostic groups (NC and MCI). They mentioned in the result section (line 236) that: "... used Kaplan-Meier analysis to determine proportion of MCI subjects with baseline CDR-SB \leq 3 (mean + 1 S.D.; 99% with global CDR of 0.5) who develop CDR-SB of 4 or more (\geq mean + 2 S.D.)" Please explain the basis of such setting and cite the corresponding literature.
- 8) In line 230-231, authors stated that: "CSF p-Tau181 and sTNFR1 levels thus appeared to have opposite relationships with cognitive decline rates in MCI". This correlation between CSF p-tau181 and cognitive decline rates in MCI patients should be discussed.
- 9) Authors analyzed 15 inflammatory proteins in CSF, but only mentioned part of them in the results of longitudinal analysis. This section should be better arranged or illustrated.

Reviewer #2 (Remarks to the Author):

In the current study, the authors investigated a set of CSF markers of microglia activity that were selected because those proteins are all shed by ADAM10/17. The authors investigated a group of demented and non-demented subjects from ADNI as well as three retrospectively analyzed samples for replication.

The authors conducted PCAs in order to determine clusters among the set of CSF markers and tested one CSF marker from each PC as a predictor of clinical progression.

The authors found in ADNI a quadratic association between higher IL-6 and global cognition in cognitive unimpaired participants and higher CSF sTNFR1 to be associated with lower cognitive decline in MCI, where the latter finding was supported by findings from a pooled analysis of two other cohorts.

Overall, the combination of different microglia related CSF biomarkers for the prediction of clinical progression and cognitive decline is of interest. However, the paper was difficult to follow due to a lack of clarity in methods description, changing outcome definitions in the Results section (especially those pertaining to predicting CDR-SB), and rudimentary characterization of the replication cohorts.

Comments

A major concern is that this paper discusses the CSF markers of microglia activation as prognostic markers for clinical AD progression. E.g. in the abstract it is stated that the authors “validated a prognostic marker for MCI in an independent cohort” (line 42). The results are interpreted such that CSF sTNFR1 and related proteins are viable biomarkers of neuroinflammation of prognosis of MCI (abstract).

The interpretation of CSF immunity markers as a prognostic markers pervades the entire manuscript. Yet no data on predictive accuracy are shown. Consequently, also no reference marker to which the the novel markers may be compared was included. sTREM2 certainly does not serve that purpose as prognostic accuracy of that marker has not been established to the best knowledge of the reviewer. There is a fine but important difference between testing 1) the association between a CSF marker of microglia activation and disease progression in order to understand the effect of microglia on disease progression , or 2) a CSF microglia marker as a prognostic marker for clinical use.

In line 126 – 127, it is mentioned that in addition to ADNI, 3 other replication cohorts were included, and the reader is referred to table 1 for sample characteristics. However, table 1 shows the sample characteristics only of that from ADNI, but none of the 3 other cohorts.

Please explain in the main text how the composite measure of cognitive decline was construed.

The rationale for first determining the PCs among biomarkers, but then test only one marker of each PC as a predictor is unclear. The fact of a correlation between particular biomarkers does not mean that they show the same function or effect on disease progression.

For example, TREM2 and progranulin were part of the same PC, but likely have opposite effects on AD pathology (which was not tested). Similarly, TNFR1 and TNFR2 share the same PC but

may show different pro vs anti-inflammatory effects.

The PCA method is not described at all. Also, please include a supplementary table on how strongly each biomarker loads on the different PCs, thus allowing the reader to assess how well the PCs segregate the different markers. For example, Rauchmann et al. (JAD, 2020, not cited in the current manuscript) who analyzed the same data in a largely overlapping sample in ADNI, showed that CSF sTNFR1 strongly correlated with TREM2 ($r > 0.6$), yet in the current paper, sTNFR1 and TREM2 were separated into two different PCs in ADNI (although they were grouped into the same PC in the other replication samples).

Lines 169 – 171: which three analytes were missing in 98 subjects? Please describe clearly for how many subjects all CSF measures were obtained. How were missing values dealt with in the statistical analysis? Perhaps a table with all the relevant information for all cohorts included would be useful.

In ADNI, the authors report that higher CSF sTNFR1 was associated with annualized decrease in CDR-SB in MCI. Does that decrease mean an improvement in cognition (which would be surprising) or an attenuated increase?

In order to validate their findings in the validation MCI cohorts the authors introduce new categorizations of clinical progression based on splitting CDR-SB scores into two categories. This is a not widely established criterion of MCI “conversion”. In order to make sure that the analysis is unbiased, please report results on the test of continuous measures of CSF sTNFR1 as a predictor of the continuous variable of CDR-SB.

The authors state that MCI in the validation cohort was due to Alzheimer pathology (line 269), yet in the methods section they state that the inclusion criterion was elevated CSF levels of p-tau (line 135).

How many LMMs were actually computed? Was the p-value corrected for multiple comparisons?

Discussion:

Role of TREM2 vs sTNFR1: the authors make far fetching conclusions about the temporal ordering of both markers, however, this is mostly speculative and no support from other studies is reported.

What do the authors think what function of TNFR1 renders the protein protective in AD?

Reviewer #3 (Remarks to the Author):

The authors measured 15 empirically derived inflammatory proteins in CSF from the ADNI and two independent cohorts, and they report a significantly lower risk of clinical progression in MCI in individuals with higher sTNFR1 concentrations. Overall, this is an interesting and well-written manuscript and I have only a few minor comments:

1. Abstract: The interpretation sentence could be re-phrased to include a more meaningful summary and outlook. It should also be mentioned how sTREM2 was associated with risk of progression (increase or decrease).
2. Discussion: The limitations of the study should be discussed in more detail.
3. Discussion: How do the authors explain that higher CSF sTREM2 was found to be related to slower cognitive outcome and less amyloid pathology in ADNI before, but apparently not in their analyses? Is it just the smaller sample size?

We have thoroughly reworked the manuscript in response to the Reviewers' comments. In particular, following Reviewer 2's recommendation, we demonstrated the full robustness of using principal component scores (instead of a representative protein's concentration) in linear mixed modeling and survival analysis. The resulting survival analysis demonstrated more dramatic outcome differences between two subgroups both having a CSF signature of Alzheimer's disease. We believe these and other modifications significantly improved the manuscript.

We include our point-by-point response to Reviewers as follows:

Reviewer #1 (Remarks to the Author):

There are substantial genetic and neuropathologic studies investigating the detrimental role of inflammation in AD, but they reported inconsistent associations of CSF inflammatory proteins with Alzheimer's core biomarkers and cognitive decline. The authors reproducibly identified four non-sTREM2 products of TACE/ADAM17 to co-vary across diagnosis, independent of core AD biomarkers. They also found increased level of CSF sTNFR1-related proteins – but not sTREM2 – to halve the risks of 36-month decline among MCI subjects with predicted Alzheimer's pathology. This research has certain scientific significance. But there remains some deficiencies. Recommendations are below:

Comment 1: The method is not well arranged. In the Ethics Approval section, authors describe: "ADNI subjects were selected to have adequate representation of healthy control subjects with normal cognition (NC; n=111) to establish normative range of inflammatory proteins ..." which should be in the Study design and participants section.

Methods: Our previously submitted manuscript was not properly formatted, and we have now revised the Methods section according to Nature Communications style.

Comment 2: In the second paragraph of the method, authors describe: "The objective of this study is to validate previous in-laboratory findings that CSF inflammatory proteins form reproducible PCs, and ..." The term PC should be given the complete spelling for it occurs at the first time. Besides, most of the content in this paragraph belongs to statistical methods rather than study design.

Response: We have now revised the Methods section according to Nature Communications style, including relocation and rearrangement of the Methods section.

Comment 3: In the third paragraph of the method, authors mentioned other cohorts independent of ADNI. These cohorts should be fully described, not merely cited. It leads to confusion whether these cohorts were from large public database or collected by the authors' team.

Response: We have now included more detailed description of the cohorts, and have included their demographic, clinical, and biomarker information in **Supplementary Table 5**.

Comment 3.1: To validate the sTNFR1 as a novel prognostic biomarker for MCI, authors chose subsets from the first and third cohorts which had rather small sample sizes (n = 33 and 16, respectively). Did the authors calculate the power of test?

Response: We apologize for the confusion created from our wording. We have now clarified the cohort selection in the Methods section. First, we included two replication cohort (Cohorts B & C, n=126 and n=68) to replicate the PCA findings from CSF biomarkers, and their demographic, clinical, and biomarker information is now included in Table S5. Second, after identifying the combination of core AD score and sTNFR1 score to associate with prognosis in the ADNI MCI cohort, we tested this finding in the second

MCI cohort consisting of MCI subjects from Cohort B (n=33) and additional subjects recruited not as part of a distinct cohort (n=16). This is better clarified in Results (page 9) and Methods-Study Design and Participants.

Comment 5: Methods of the cognition assessment should be described in detail, especially the cognitive z-score.

Response: We have now provided further detail on cognitive assessment in the ADNI cohort (page 17). In ADNI, composite memory (Mem) and executive function (EF) scores were previously generated by Gibbons, Crane, and co-workers using traditional neuropsychological measures through item-response theory and bi-factor confirmatory factor analysis. ADNI-Mem and ADNI-EF were both normed to have mean of 0 and standard deviation of 1, which is equivalent to Z-transformation which led us to previously use the Z-score terminology. To be clear, we have now specified that we averaged the ADNI-Mem and ADNI-EF scores to reflect overall cognition, and we have now re-termed the entity ADNI-Mem-EF.

Comment 6: The inclusion and exclusion criteria of participants enrolled in the study should be stated clearly in the method section.

Response: We have now provided additional information on the two additional cohorts in Methods-Study design and participants, and inclusion/exclusion criteria in Supplementary Material.

Comment 7: Authors should better describe the prognostic outcome indicators for different diagnostic groups (NC and MCI). They mentioned in the result section (line 236) that: "... used Kaplan-Meier analysis to determine proportion of MCI subjects with baseline CDR-SB ≤ 3 (mean + 1 S.D.; 99% with global CDR of 0.5) who develop CDR-SB of 4 or more (\geq mean + 2 S.D.)" Please explain the basis of such setting and cite the corresponding literature.

Response: We had previously devised these thresholds as the ADNI cohort had overlapping CDR-SB scores according to diagnosis (now demonstrated in Figure S2), with CDR-SB threshold of 3 providing the most optimal separation between MCI and dementia diagnosis. This is consistent with the normative data from the Knights Alzheimer's Disease Research Center which we have now included as an additional citation (45). Our findings remained consistent whether we analyzed the proportion of MCI subjects who converted to dementia diagnosis or CDR-SB ≥ 4 . Therefore, we have now presented this data in the figures and tables to minimize confusion.

Comment 8: In line 230-231, authors stated that:" CSF p-Tau181 and sTNFR1 levels thus appeared to have opposite relationships with cognitive decline rates in MCI". This correlation between CSF p-tau181 and cognitive decline rates in MCI patients should be discussed.

Response: We have now elaborated on this in two places. First, we emphasized the relationship between faster decline and higher core AD score (page 6). Second, we transformed high core AD score into a corresponding p-Tau₁₈₁ level, and elaborated on the relationship between p-Tau181 levels and rates of cognitive decline (page 8).

Comment 9: Authors analyzed 15 inflammatory proteins in CSF, but only mentioned part of them in the results of longitudinal analysis. This section should be better arranged or illustrated.

Response: We used PCA as a dimension reduction approach to analyze the fewest number of independent variables. Our analytical approach using linear mixed modeling (LMM) began with a base model using demographic variables, APOE $\epsilon 4$, core AD score, and time. Following this, each PC score

was tested in a stepwise fashion to arrive at a final model which improved moderately or substantially improved the Aikake Information Criterion. This is more now explicitly stated in Methods-Statistical Analysis (page 18).

Reviewer #2

In the current study, the authors investigated a set of CSF markers of microglia activity that were selected because those proteins are all shed by ADAM10/17. The authors investigated a group of demented and non-demented subjects from ADNI as well as three retrospectively analyzed samples for replication.

The authors conducted PCAs in order to determine clusters among the set of CSF markers and tested one CSF marker from each PC as a predictor of clinical progression.

The authors found in ADNI a quadratic association between higher IL-6 and global cognition in cognitively unimpaired participants and higher CSF sTNFR1 to be associated with lower cognitive decline in MCI, where the latter finding was supported by findings from a pooled analysis of two other cohorts.

Comment: Overall, the combination of different microglia related CSF biomarkers for the prediction of clinical progression and cognitive decline is of interest. However, the paper was difficult to follow due to a lack of clarity in methods description, changing outcome definitions in the Results section (especially those pertaining to predicting CDR-SB), and rudimentary characterization of the replication cohorts.

Response: We have now followed the recommendations of Reviewers 1 & 2 to provide additional details and clarity to Methods, aligned the outcome measures in Results, and provided additional information on the replication cohorts.

Comments: A major concern is that this paper discusses the CSF markers of microglia activation as prognostic markers for clinical AD progression. E.g. in the abstract it is stated that the authors “validated a prognostic marker for MCI in an independent cohort” (line 42). The results are interpreted such that CSF sTNFR1 and related proteins are viable biomarkers of neuroinflammation of prognosis of MCI (abstract). The interpretation of CSF immunity markers as a prognostic markers pervades the entire manuscript. Yet no data on predictive accuracy are shown. Consequently, also no reference marker to which the novel markers may be compared was included.

Response: We closely followed the definition of **prognostic** biomarker by FDA-NIH Biomarker Working Group outlined in Biomarker, EndpointS, and other Tools (BEST) which defined it as “A biomarker used to identify likelihood of a clinical event, disease recurrence or progression in patients who have the disease or medical condition of interest” which is now clarified and cited (page 10). The survival analysis we used is in keeping with the likelihood framework, and we presented the appropriate odds ratios associated with the likelihood framework. A **predictive** biomarker, on the other hand, is defined as one “used to identify individuals who are more likely than similar individuals without the biomarker to experience a favorable or unfavorable effect from exposure to a medical product or an environmental agent.” Therefore, we do not find predictive accuracy appropriate for this prognostic marker work. In terms of reference markers, we consistently used in our models the reference model of including age, sex, *APOE* genotype, and core AD score as the reference model, with the improved model additionally incorporating sTNFR1 PC score. We used the change in AIC value to support the appropriateness of including one additional variable.

Comment: In line 126 – 127, it is mentioned that in addition to ADNI, 3 other replication cohorts were included, and the reader is referred to table 1 for sample characteristics. However, table 1 shows the sample characteristics only of that from ADNI, but none of the 3 other cohorts.

Response: As above, we have now provided additional information on Cohorts B & C (replication cohort for PCA) in **Supplementary Table 5**. The MCI progression replication cohort's characteristics are in Table 4.

Comment: Please explain in the main text how the composite measure of cognitive decline was construed.

Response: This is now clarified in the Methods section.

Comment: The rationale for first determining the PCs among biomarkers, but then test only one marker of each PC as a predictor is unclear. The fact of a correlation between particular biomarkers does not mean that they show the same function or effect on disease progression.

Response: We appreciate this comment from Reviewer 2. We had previously chosen to not present the PC-based model to reach the broadest readership. Upon this recommendation by Reviewer 2, we have since revised the manuscript to use the PC scores in the longitudinal progression models. This has created much more divergent conversion rates between the biomarker-defined groups (Figs 1B & 2A), although we do also include more reader-friendly versions using protein concentrations (Figs 2C and 4). For the sTNFR1 score, we also included the linear regression formula incorporating concentrations of sTNFR1, sTNFR2, and sVCAM1.

Comment: For example, TREM2 and progranulin were part of the same PC, but likely have opposite effects on AD pathology (which was not tested). Similarly, TNFR1 and TNFR2 share the same PC but may show different pro vs anti-inflammatory effects.

Response: We respectfully disagree with the Reviewer's interpretation that proteins with different biological effects would necessarily belong to different PCs. In fact, here we have shown empirically that proteins expected to have opposite effects *in vitro* (eg, IL-6 as "pro-inflammatory" and IL-10 as "anti-inflammatory") are highly correlated with each other at the bulk fluid level. At the tissue level, many of these proteins have receptors on immune cells with different physiologic and pathologic roles. Cytokine receptors are additionally present on neurons which are neither pro- nor anti-inflammatory. Through the current work, we hope to change other investigators' view that markers with certain *in vitro* or cellular functions uniformly correspond to a single *in vivo* biological events.

Comment: The PCA method is not described at all. Also, please include a supplementary table on how strongly each biomarker loads on the different PCs, thus allowing the reader to assess how well the PCs segregate the different markers.

Response: We have now provided additional details for the PCA. PCA was first conducted using only subjects with all proteins measures (n=241), and then with missing values replaced with mean. Loading ≥ 0.400 using both methods were forwarded to Table 2 (n example for MCI is now given in Table S4). The variable loading was further confirmed by PCA with imputation through expectation maximization. This is now included in a new Missing Data information.

Comment: For example, Rauchmann et al. (JAD, 2020, not cited in the current manuscript) who analyzed the same data in a largely overlapping sample in ADNI, showed that CSF sTNFR1 strongly correlated with TREM2 ($r > 0.6$), yet in the current paper, sTNFR1 and TREM2 were separated into two different PCs in ADNI (although they were grouped into the same PC in the other replication samples).

Response: We submitted this manuscript for initial review before the publication of Rauchmann et al which uses the protein levels we measured and uploaded into ADNI, but we have now cited the article. In terms of protein correlations, we do not deny that variables can load onto multiple PCs with variable loading strength. We follow the common practice of selecting a loading threshold 0.400 to determine

factors loading most strongly onto each PC, but we acknowledge the difference in loading scores of 0.401 and 0.399 is arbitrary. However, we bring the Reviewers' and readers' attention to the very strong loading of sTNFR1 score by sTNFR2, sTNFR2, and sVCAM1, vs more modest loading of one sTREM2 assay result in one model. We also note that sTREM2 loading onto an independent PC better explained the variance, which PCA allowed us to discover beyond correlational analysis.

Comment: Lines 169 – 171: which three analytes were missing in 98 subjects? Please describe clearly for how many subjects all CSF measures were obtained. How were missing values dealt with in the statistical analysis? Perhaps a table with all the relevant information for all cohorts included would be useful.

Response: This is now explained in detail in Missing data.

Comment: In ADNI, the authors report that higher CSF sTNFR1 was associated with annualized decrease in CDR-SB in MCI. Does that decrease mean an improvement in cognition (which would be surprising) or an attenuated increase?

Response: The decrease in annualized change in CDR-SB is most appropriately interpreted as an attenuated increase, but we have removed this sentence in the Revised manuscript for clarity.

Comment: In order to validate their findings in the validation MCI cohorts the authors introduce new categorizations of clinical progression based on splitting CDR-SB scores into two categories. This is a not widely established criterion of MCI "conversion". In order to make sure that the analysis is unbiased, please report results on the test of continuous measures of CSF sTNFR1 as a predictor of the continuous variable of CDR-SB.

Response: We have included the effect of sTNFR1 score on continuous variables of ADNI-Mem-EF and CDR-SB in the new Table 3. As in our response to Reviewer 1, we have also now included conversion using the more conventional nomenclature (MCI to dementia).

Comment: The authors state that MCI in the validation cohort was due to Alzheimer pathology (line 269), yet in the methods section they state that the inclusion criterion was elevated CSF levels of p-tau (line 135).

Response: We have introduced the term predicted Alzheimer's disease neuropathologic change (predicted ADNC) corresponding to $t\text{-Tau}/A\beta_{42} \geq 0.39$. Using the correlation between core AD score, $t\text{-Tau}/A\beta_{42}$, and $p\text{-Tau}_{181}$, we first derived the core AD score threshold associated with predicted ADNC (page 6), and then the $p\text{-Tau}_{181}$ levels associated with this core AD score (page 8).

Comment: How many LMMs were actually computed? Was the p-value corrected for multiple comparisons?

Response: For each diagnosis, a single LMM is first conducted with only demographic variables, APOE $\epsilon 4$ allele status, and core AD score (page 17). A second LMM was run for the stepwise introduction of additional PC scores. Regression models generally do not need to account for multiple comparisons within the same model, and we do not consider experiment-wide adjustment necessary as we divided the ADNI cohort according to diagnosis. We also did not adjust for multiple comparisons between the two outcome measures (ADNI-Mem-EF and CDR-SB) because the model using CDR-SB was to demonstrate consistency of results, not to replicate results or identify new markers.

Comment: Role of TREM2 vs sTNFR1: the authors make far fetching conclusions about the temporal

ordering of both markers, however, this is mostly speculative and no support from other studies is reported.

Response: We disagree with the Reviewer's portrayal of our discussion as far fetching. We make it a habit to restrict the degree of speculation in the writing of our Discussion, and the temporal ordering of biomarkers is no more speculative than the pro-/anti-inflammatory role of these biomarkers. We merely suggested that, based on our observation of sTNFR1-related proteins and sTREM2 proteins associating with prognosis at different stages of AD, temporal ordering may be one explanation. However, we have toned down this interpretation in the revision.

Comment: What do the authors think what function of TNFR1 renders the protein protective in AD?

Response: We did not speculate anything about the role of TNFR1 in being protective in AD, nor do we suggest sTREM2 to be protective in AD. We consider it premature to enter into therapeutic development with aims of boosting either, and we have emphasized this in our discussion (page 11) - as such an assumption would have led to amyloid-boosting therapy in Alzheimer's disease.

Reviewer #3 (Remarks to the Author):

The authors measured 15 empirically derived inflammatory proteins in CSF from the ADNI and two independent cohorts, and they report a significantly lower risk of clinical progression in MCI in individuals with higher sTNFR1 concentrations. Overall, this is an interesting and well-written manuscript and I have only a few minor comments:

Response: We thank Reviewer 3 for these positive comments.

Comment: Abstract: The interpretation sentence could be re-phrased to include a more meaningful summary and outlook. It should also be mentioned how sTREM2 was associated with risk of progression (increase or decrease).

Response: We have now rephased it as "higher soluble TREM2 levels only associated with slower decline in the dementia stage of AD. "

Comment: Discussion: The limitations of the study should be discussed in more detail.

Response: We have now included additional limitations in Discussion, including:

- Page 11: these loading profiles should still be prospectively tested using drugs known to alter one or more of the proteins.
- Page 12: the selective nature of the ADNI cohort, the smaller sample size of the Atlanta MCI cohort, not correlating with measures of brain atrophy or ischemia on MRI, and not having adequate power in longitudinal analysis to examine the effect of race on biomarker-based prognosis.

Comment: Discussion: How do the authors explain that higher CSF sTREM2 was found to be related to slower cognitive outcome and less amyloid pathology in ADNI before, but apparently not in their analyses? Is it just the smaller sample size?

Response: We also found higher CSF sTREM2 to relate to slower cognitive decline in the AD dementia group, but not the MCI group. Ewers and colleagues (EMBO Molecular Medicine, 2020) found higher CSF sTREM2 levels associated with slower amyloid plaque deposition by PET imaging, although there was significant overlap between those with high vs. low sTREM2 levels. Ewers and colleagues also reported the effect of sTREM2 on cognitive decline in ADNI, but grouped MCI subjects with those with dementia. We believe this grouping accentuated the significance of CSF sTREM2 levels across the cohort, especially in the absence of a better biomarker (eg, sTNFR1).

REVIEWER COMMENTS

Reviewer #1 (Remarks to the Author):

This study does have certain limitations, including the selective nature of the ADNI cohort, and the small sample size of the MCI cohort which might reduce the power in longitudinal analyses of biomarker-based prognosis. The authors have mentioned these deficiencies in the limitation section of the revised manuscript.

Reviewer #2 (Remarks to the Author):

The authors have adequately responded to all comments. The reference list need to be updated though and extended by all references discussed in the rebuttal letter.

Reviewer #3 (Remarks to the Author):

The authors have fully addressed my concerns. I have no further comments or suggestions.

Reviewer #1 (Remarks to the Author):

This study does have certain limitations, including the selective nature of the ADNI cohort, and the small sample size of the MCI cohort which might reduce the power in longitudinal analyses of biomarker-based prognosis. The authors have mentioned these deficiencies in the limitation section of the revised manuscript.

Response: We agree with the Reviewer on these limitations.

Reviewer #2 (Remarks to the Author):

The authors have adequately responded to all comments. The reference list need to be updated though and extended by all references discussed in the rebuttal letter.

Response: We only located one reference which was in the Rebuttal Letter but not in the revised manuscript (reference 27) which was deleted by accident. We have now added this reference back.

Reviewer #3 (Remarks to the Author):

The authors have fully addressed my concerns. I have no further comments or suggestions.